# Spaceborne LiDAR Systems: Evolution, Capabilities, and Challenges

**DOI:** 10.3390/s25123696

**Published:** 2025-06-12

**Authors:** Jan Bolcek, Mohamed Barakat A. Gibril, Jiří Veverka, Šimon Sloboda, Roman Maršálek, Tomáš Götthans

**Affiliations:** 1Department of Radio Electronics, Faculty of Electrical Engineering and Communication, Brno University of Technology, 602 00 Brno, Czech Republic; 205866@vut.cz (J.V.); 203423@vut.cz (Š.S.); marsaler@vutbr.cz (R.M.); gotthans@vut.cz (T.G.); 2GIS and Remote Sensing Center, Research Institute of Sciences and Engineering, University of Sharjah, Sharjah 27272, United Arab Emirates; mbgibril@sharjah.ac.ae

**Keywords:** LiDAR, spaceborne LiDAR, remote sensing, earth observation, planetary exploration, atmospheric studies, LiDAR technology, topographic mapping, climate change monitoring, laser altimetry, space missions

## Abstract

In the realm of earth observation and space exploration, LiDAR technology offers humanity insights into the dynamics of our planet and beyond. This paper reviews spaceborne LiDAR instruments with attention to their evolution, capabilities, and achievements. We focus on the high-level LiDAR instrument design, their components, and their operational parameters in contribution to the study of Earth. Through examining selected space missions, this work illustrates the role of LiDAR technology in our understanding of environmental and atmospheric phenomena. Furthermore, the paper looks ahead, discussing the ongoing development of advanced LiDAR technologies.

## 1. Introduction

LiDAR(Light Detection and Ranging) is an active remote sensing technique that has advanced our capability to observe and measure the Earth’s surface, atmosphere, and planetary science [1,2]. Since its inception in the 1960s, shortly after the discovery of lasers, LiDAR technology has evolved considerably from ground-based systems to sophisticated spaceborne instruments capable of taking detailed measurements of Earth’s topography, vegetation, and atmospheric constituents [3,4].

LiDAR systems for terrestrial applications can be broadly classified based on their deployment platforms into three distinct categories: (1) ground-based LiDAR, (2) airborne LiDAR, and (3) spaceborne LiDAR [5]. Initially developed in the 1960s [6], ground-based LiDAR systems pioneered the application of laser technology for precise distance measurements and atmospheric studies. These early systems laid the foundation for atmospheric studies, including measurements of cloud height and monitoring of air pollution [7]. Modern ground-based LiDAR systems have evolved to include scanning capabilities, enabling the creation of high-resolution three-dimensional models of structures, urban landscapes, and vegetation. These systems are widely used in geotechnical monitoring, civil engineering, and environmental research due to their ability to provide highly precise spatially localized measurements [7,8]. Later, in the 1970s and 1980s, LiDAR technology extended its range and applications through integration with airborne platforms [9]. These airborne systems provided greater flexibility and coverage, enabling topographic mapping of large and inaccessible areas by generating accurate Digital Elevation Models (DEMs) by providing high-density ground point measurements, enabling the creation of accurate and detailed elevation grids in accordance with established geospatial definitions [10]. DEMs themselves are essential tools in various fields including hydrological studies, urban planning, disaster risk assessment, and archaeology [10,11,12,13]. In addition, airborne LiDAR became an asset in forest applications such as quantifying canopy height, biomass, and forest structure and helped to understand carbon sequestration and biodiversity [14,15]. Lastly, airborne LiDAR transformed archaeological explorations, applications of airborne LiDAR have revealed discovering previously hidden structures beneath dense vegetation, such as Mayan city layouts in Central America [12].

The need for global-scale observations with consistent accuracy and resolution drove the transition towards spaceborne LiDAR, where LiDAR serves as a payload on the satellite/spacecraft to provide continuum measurements [2]. A critical milestone was the deployment of the first spaceborne LiDAR system called LiDAR In-Space Technology Experiment (LITE). In the 1990s [16], LITE and subsequent spaceborne LiDAR missions contributed to our understanding of Earth’s topography, ice sheet dynamics, and forest canopy structure and provided data for a wide range of scientific, environmental and commercial applications, such as climate change research and environmental monitoring [2,7,17,18]. Figure 1 provides an overview of important milestones of the development of LiDAR technology.

Beyond its terrestrial applications, LiDAR has also played an increasingly important role in planetary science, commencing with the Apollo 15 mission in 1971, where a laser altimeter—a precursor to modern LiDAR systems—mapped the Moon’s surface and provided valuable topographic data [22,23].

This paper comprehensively reviews spaceborne LiDAR technology, focusing on its evolution, underlying principles, applications, and future prospects, particularly in terrestrial applications and contributions to Earth observation. Section 2 describes the fundamental working principles of LiDAR systems, detailing their key components, the LiDAR equation, and the system characterization. Section 3 categorizes LiDAR system types based on measurement techniques, such as backscatter LiDAR, differential absorption LiDAR (DIAL), Doppler wind LiDAR, and laser altimetry. Section 4 describes the challenges and limitations of spaceborne LiDAR missions, such as spatial resolution trade-offs. Section 5 provides an overview of past and current space missions that have used LiDAR technology, such as LITE, ICESat, CALIPSO, and GEDI. Section 6 discusses the applications and studies based on the data derived from spaceborne LiDARs in various fields, such as atmospheric aerosol profiling, vegetation structure mapping, cryosphere monitoring, and urban studies. Finally, Section 7 explores the future directions of spaceborne LiDARs, including planned missions and next-generation concepts.

## 2. Basic Principle and Functionality

LiDAR, also called LaDAR or optical radar [24], employs electromagnetic waves in the optical spectrum to measure the distance between the sensing instrument and a target object time. Additionally, by analyzing the interaction of the emitted radiation with the target—such as scattering, absorption, reflection, and fluorescence—one can also infer its physical characteristics [24,25]. As an active remote sensing technique, LiDAR provides its own illumination source, enabling data acquisition both during the day and at night, including regions of total darkness, such as the prolonged darkness of polar areas [26,27].

### 2.1. Instrument Configuration and Signal Detection

Figure 2 illustrates the high-level setup schematic of the spaceborne LiDAR instrument which consists of three main subsystems: transmitter, receiver, and control electronics + thermal control. The transmitter includes a laser light source and optical elements (such as mirrors, lenses, or beam expanders) used to shape, collimate, or split the beam. The optical receiver consists of a telescope that collects the backscattered photons from the target, spatial and spectral filters that select the desired wavelengths, and a photodetector. The designs and choice of these elements vary among systems and depend on the application. If the transmitter and receiver share optical elements, the setup is called monostatic; if they have separate paths, it is bistatic. For example, the CALIOP instrument onboard the CALIPSO satellite is a bistatic LiDAR, using two lasers aligned parallel to the receiving telescope [24,28].

#### 2.1.1. Transmitter Subsystem

The transmitter in spaceborne LiDAR system typically comprises a laser system source integrated with optical elements components [24]. Commonly employed lasers include solid-state neodymium-doped yttrium aluminum garnet (Nd:YAG) or neodymium-doped yttrium orthovanadate (Nd:YVO4) lasers accompanied by the Q-Switching technique to produce near-infrared (NIR) pulses at 1064 nm. These laser outputs can be frequency-doubled or tripled to produce emissions at 532 nm (visible) and 355 nm (ultraviolet), thereby extending their application to a broader range of Earth and atmospheric observations [29,30].

The choice of laser wavelength notably influences their performance and applicability. The 1064 nm wavelength, common in aerial and spaceborne mapping, benefits from widely available laser sources, affordable silicon-based detectors, and strong reflectivity from typical terrestrial surfaces such as vegetation, snow, and soil [29]. However, it is less suitable and effective in aquatic environments due to increased absorption by water molecules, which limits its suitability for bathymetric applications [31]. Furthermore, eye safety constraints can be more stringent at 1064 nm compared to some shorter wavelengths, requiring careful system design for orbital or aircraft-based instruments [32]. In contrast, the 532 nm wavelength penetrates water molecules [31,33] rendering it useful for bathymetric LiDAR measurements. However, in the visible spectrum, 532 nm systems can be affected by ambient light, requiring filtering techniques to improve signal-to-noise ratios [34]. The shorter wavelength increases Rayleigh scattering; therefore, the 355 nm wavelength allows highly sensitive detection of aerosols and trace gases [35].

Q-switching is a laser technique that produces short, intense pulses by temporarily storing energy in the laser medium and then rapidly releasing it. This is achieved by controlling the laser cavity’s quality factor (Q), which measures energy efficiency. Q-switching enables the generation of nanosecond-range pulses with peak powers much higher than in continuous operation [36].

The laser excitation in LiDAR systems is typically achieved through optical pumping, using either flash lamps or laser diodes [37]. This process supplies energy to the laser medium to achieve population inversion, allowing electrons to transition to higher energy states and subsequently release photons during relaxation—thereby sustaining coherent laser emission [38].

Recent advances in spaceborne LiDAR technology have focused on improving the efficiency, reliability, and compactness of laser transmitters. For instance, NASA’s Goddard Space Flight Center has been advancing laser sources for space-based LiDAR and communication applications, including diode-pumped solid-state lasers (DPSSLs) and fiber-based laser technologies [39]. Additionally, advances in passively Q-switched lasers using silicon photonics have shown promise for compact and efficient laser systems suitable for spaceborne LiDAR [40].

The key high-level parameters to describe the LiDAR laser system can include the following [18,21,41,42,43,44,45,46]:Laser wavelength [nm]: Commonly selected based on the application’s sensing requirements.Pulse repetition frequency (PRF) [Hz]: PRF refers to the number of laser pulses emitted per second. This parameter directly influences spatial resolution and ground sampling density. Although higher PRFs enable denser sampling, they also increase power demand and thermal load. Spaceborne systems typically operate at up to 20 kHz.Laser pulse energy [mJ]: Pulse energy determines the system’s ability to detect weak returns from distant or low-reflectance surfaces. Higher energies support longer-range and optically complex measurements (e.g., dense clouds or thick vegetation canopies) but require greater power and thermal control. Typical values for spaceborne LiDARs range from 1 to 100 mJ.No. of laser beams [-]: The number of laser beams affects swath width, spatial resolution, and data redundancy. While early missions such as LITE and GLAS employed single-beam configurations, more recent systems such as ATLAS and GEDI utilised multiple beams—typically six or eight—to enhance coverage and efficiency.

#### 2.1.2. Receiver Subsystem

The receiver typically consists of a telescope equipped with spatial and spectral filters and an electronic photodetector [24]. Receiver sensitivity is a primary concern, as transmitted laser power can be on the order of megawatts (MW), while the received power is typically reduced to nanowatts (nW). Furthermore, the sensitivity of the photodetector strongly correlates with the overall performance of the system, directly affecting the capability [47].

LiDAR receivers generally operate in either analog (linear/full-waveform) mode or digital (photon-counting) mode [48]. Photodetectors commonly used in LiDAR systems include photomultiplier tubes (PMTs) and avalanche photodiodes (APDs) [48,49,50], both of which can function in either analog or digital mode. When operated in Geiger mode (digital mode), APDs are referred to as Single-Photon Avalanche Diodes (SPADs) [48]. The choice of detector type depends on the mission requirements and the wavelength range of interest. PMTs have historically been preferred for ultraviolet (UV) and visible wavelengths, such as 355 nm and 532 nm, due to their high gain and reasonable quantum efficiency at those wavelengths [48,51]. On the other hand, APD offers better performance at the 1064 nm, where PMTs exhibit poor quantum efficiency [51].

Photon counting sensors are capable of registering individual photon events, allowing for high vertical resolution, enhanced sensitivity, and reduced susceptibility to electronic noise, including dark counts and analog-to-digital converter (ADC) noise [30,52]. In addition, they often provide higher signal-to-noise ratios (SNRs) for a given signal energy. However, a key limitation of photon-counting detectors is dead time, which refers to an interval following a photon detection during which the sensor cannot register another event. This reset period, which typically lasts tens of nanoseconds, restricts the maximum achievable count rate and can cause a signal loss in high-flow environments [30,53].

Several techniques have been developed to mitigate dead-time effects, such as the following: (1) using low photon rates and (2) using an array of detectors to reduce the chance that multiple photons are hitting the same detector at the same time [30]. Various examples can include multidetector arrays and multichannel signal preprocessing circuits that have been developed to independently preprocess each detection channel, significantly reducing noise counts and suppressing dead-time effects [54]. Furthermore, advanced dead-time correction algorithms have been proposed to improve the accuracy and efficiency of photon-counting LiDAR systems [55,56].

Expanding on photon-counting capabilities, SPAD arrays have been developed with a smaller pitch, higher pixel density, and integrated processing electronics, enabling better resolution and accuracy in 3D imaging [57]. Unlike single-element SPADs, Silicon Photomultipliers (SiPMs) integrate multiple avalanche diodes to form a high-efficiency photon detection device. SiPMs are promising candidates for future LiDAR missions. SiPMs are arrays of SPADs connected in parallel, acting as a solid-state alternative to PMTs [58]. They are known for their high photon detection efficiency and low noise characteristics. They are instrumental in applications requiring high sensitivity and low light detection. Recent studies have shown that SiPMs can maintain their performance even in harsh space environments. For example, GRBAlpha and VZLUSAT-2 CubeSats have demonstrated the durability of SiPM arrays over three years of in-orbit operation, highlighting their potential for long-term space missions [59].

A review of recent missions reveals how the aforementioned preferences and considerations have translated into real-world detector configurations. For instance, the LITE mission utilized PMT detectors operated in analog mode on their 532 nm and 355 nm channels. At the same time, CALIOP used a PMT operated in analog mode on both of their 532 nm channels [16,51]. In contrast, analog mode silicon APDs have been used for the 1064 nm channels where high sensitivity in the near-IR is needed. The GLAS instrument in ICESat (2003) and CALIPSO’s CALIOP (2006) employed silicon APDs for their 1064 nm return [51,60]. Regarding PMTs, the Atlas instrument from the ICESat-2 mission utilized an array of PMTs with 16 individual photon multiplier channels for a single-photon counting detection [61].

The key high-level parameters to describe the receiver subsystem can include [53,61,62,63,64]:Field of view (FOV) [μrad]: The FOV is the angular range over which the LiDAR system can detect backscattered light and is typically in the range of (100–1000) μrad. A wider FOV can capture more scattered light but may also increase background noise, affecting the signal-to-noise ratio (SNR).Quantum efficiency (QE) [%]: Probability that an incident photon generates a photoelectron.Photon detection efficiency (PDE) [%] is a variable that describes the probability that a photon will be detected and is mainly dependent on the quantum efficiency of the detector semiconductor material and the arrangement of the sensors.Dead time [ns] is the period immediately after detecting a photon during which the detector cannot register another photon. A short dead time allows the detector to be ready to detect another photon more quickly, enhancing the counting rate and efficiency.Timing jitter [ps]: A low jitter is essential for applications requiring precise timing measurements, such as time-correlated single photon counting (TCSPC). Jitter refers to the variability in timing accuracy when detecting photons. Reducing jitter improves the temporal resolution of measurements, which is crucial for accurately determining the time of arrival of photons.Dark count rate [counts per second]: The dark count rate measures the number of false counts detected by the sensor, essentially background noise. Minimizing this rate leads to achieving high signal-to-noise ratios in sensitive applications, allowing for detecting very low levels of light without significant interference from the detector itself.

### 2.2. LiDAR Equation

The signal received by a LiDAR system carries information about various properties of the target, including distance, reflectivity, and motion. A theoretical framework for understanding and quantifying these measurements is provided by the LiDAR equation. It relates the received return signal to the characteristics of the transmitted pulse, the propagation medium, and the properties of the target. The general form of the LiDAR equation is [7,65,66]:(1)P(R)=KG(R)β(R)T(R)
where

P(R) is the power received from a distance *R*.*K* is a constant factor.G(R) describes the geometric spreading (like $\frac{1}{R^{2}}$ fall-off due to spreading loss).β(R) is the backscatter coefficient at a distance *R*.T(R) propagation medium transmission factor.

The parameters *K* and G(R) are determined by the LiDAR system configuration, including transmitter power and overall system efficiency. The backscatter coefficient β(R) characterizes the target’s ability to scatter light back toward the receiver, while T(R) accounts for transmission losses due to absorption and scattering along the beam path. These terms can be further specified based on particular LiDAR systems and applications [24,66].

### 2.3. Key System Parameters

A spaceborne LiDAR system is characterized by several high-level performance metrics that influence its resolution, coverage, and overall data quality [26,30,41,42,45,67,68]:Footprint diameter: The footprint diameter is the size of the area on Earth’s surface that a single LiDAR pulse illuminates and measures. The footprint size influences the spatial resolution and the ability to detect fine-scale features on the Earth’s surface.Horizontal resolution: The horizontal resolution is the smallest resolvable distance between two footprints, and is typical;y in the range of hundreds of meters. The higher spatial resolution allows for more detailed mapping and analysis of surface features.Vertical resolution: The vertical resolution is the smallest resolvable distance in the vertical direction. It depends on the application: hundreds of meters for atmospheric detection (aerosol/cloud layers) and tens of centimeters for altimetry (surface elevation and structure measurements). The improved vertical resolution improves the ability to profile atmospheric layers and surface topography.Accuracy: The accuracy of spaceborne LiDAR measurements depends on various factors, including the calibration of the LiDAR system, atmospheric conditions, and surface reflectance properties. Accurate calibration and correction for atmospheric effects are crucial for reliable measurements.Scanning pattern: The scanning pattern is the pattern in which the LiDAR system scans the ground and can be a raster or a swath pattern. The choice of scan pattern affects the coverage area and the density of data points collected.The LiDAR coverage/swath width: The swath width determines the LiDAR coverage. A wider swath width increases the coverage area but may reduce spatial resolution.Signal-to-noise ratio (SNR): SNR is critical to determining the detection capability and precision of a LiDAR system. Higher SNRs indicate more reliable and precise measurements, influencing the overall quality of the data products.

Figure 3 includes the dimensioned parameters for high-level description of spaceborne LiDAR instrument.

## 3. Types of LiDARs

LiDAR instruments can be classified based on several criteria. A common classification was mentioned earlier in the introduction—based on the deployment platform. However, for the purposes of this paper, LiDAR systems are categorized on the basis of their light-target interaction phenomena. Based on this criterion, LiDAR instruments can be classified as follows [24,69]:Atmospheric backscattering LiDARs.Differential absorption LiDARs (DIALs).Doppler (wind) LiDARs.Ranging and altimeter LiDARs.Full-waveform LiDARs.

### 3.1. Atmospheric Backscattering LiDAR

Scattering is a physical phenomenon that refers to the redirection of electromagnetic waves when they encounter particles or irregularities in their path. Backscattering, in particular, occurs when a portion of the scattered light is redirected back toward the source due to interactions with atmospheric molecules or aerosols. The intensity and nature of backscattering depend on a couple of factors, such as the wavelength of the laser, the size and composition of atmospheric particles, and the distance the laser pulse travels through the medium. An illustration of the scattering phenomenon is depicted in Figure 4 [66,70,71,72].

In the context of LiDAR, atmospheric backscattering LiDARs are used to analyze the atmosphere of the Earth, mainly to study aerosols, clouds, and atmospheric gases [16,73].

The fundamental working principle of atmospheric backscattering LiDAR involves the following: (1) emission of laser pulses into the atmosphere, (2) backscattering of a fraction of the laser light due to interactions with atmospheric particles or gas molecules, and (3) detection and analysis of the signal returned to derive atmospheric properties [7,70,74,75]. The backscatter coefficient (β) quantifies the signal strength. For spaceborne LiDARs, this coefficient is measured as a function of altitude [7,76].

### 3.2. Types of Atmospheric Scattering

Atmospheric scattering in the atmosphere occurs in two primary modes [77]:Elastic scattering —The wavelength of the scattered light remains unchanged. Elastic LiDAR does not detect specific chemicals. Instead, it measures how different gases, particles, and aerosols scatter light. This helps identify areas where the atmosphere changes, such as differences in density, humidity, dust, and pollution [70,74,78].Inelastic scattering—The wavelength shifts due to energy exchanges with molecules (e.g., “Raman scattering”) Raman scattering measures a spectral response and can monitor multiple gas species simultaneously [79,80,81].

There are two main types of elastic scatterings in the atmosphere of the Earth:Mie scattering occurs when particle sizes are similar to or larger than the wavelength of light (e.g., dust, water droplets, aerosols, molecules $NO_{2}$ and $O_{2}$) and scale with the $d^{2}$ where *d* is the diameter of the particle. It is dominant in the lower atmosphere, where larger particles are present [72,82,83,84].Rayleigh scattering is the scattering of light by particles much smaller than the wavelength of the light (e.g., dust, pollen, smoke, and water vapor). The intensity of Rayleigh scattering is inversely proportional to the fourth power of the wavelength with factor $\lambda^{-4}$. This means that shorter wavelengths (such as 355 nm) are scattered much more strongly than longer wavelengths (such as 1064 nm) [35] and are more predominant in the upper parts of the atmosphere [72,82,83,84].

Another essential phenomenon related to the atmospheric LiDAR system is depolarization, which occurs when the polarization state of light changes as it interacts with matter, such as aerosols or clouds in the atmosphere. In spaceborne LiDARs, depolarization is an important parameter that can provide information about the properties of the target being studied. For example, the linear depolarization ratio of cirrus clouds mainly reflects the single-scattering properties of non-spherical ice particles. Similarly, depolarization data can help distinguish between different aerosols or clouds in the atmosphere. Notably, the depolarization ratio can be used to identify non-spherical dust particles that unambiguously quantify dust in the atmosphere and improve the accuracy of LiDAR measurements by correcting for depolarization effects [85,86,87].

### 3.3. HSRL—High-Spectral-Resolution LiDAR

High-spectral-resolution LiDAR (HSRL) is a specialized atmospheric system designed to distinguish between aerosol (Mie) and molecular (Rayleigh) scattering based on their spectral signatures. By isolating these components, HSRL enables direct retrieval of aerosol optical depth (AOD)—a key parameter for quantifying aerosol extinction effects [88,89]. This separation is achieved using a high-spectral-resolution filter, which allows measurements of aerosol extinction and backscatter independently of molecular scattering [90]. As such, the capability is particularly useful for studying the vertical distribution of aerosols and their optical properties, as well as for improving the accuracy of aerosol measurements in the presence of clouds [88].

### 3.4. Diferential Absorption LiDARs

Differential absorption LiDAR (DIAL) is designed to detect and identify the atmospheric or surface constituents (such as ozone or water vapor) with high precision and vertical resolution throughout the troposphere and lower stratosphere [91,92]. The operational principle of DIALs is as follows: two laser pulses with different wavelengths are emitted, where one wavelength is absorbed by the investigated substance more strongly than the other. The differential molecular absorption coefficient ($\alpha_{mol,abs}$) can be determined by comparing the absorption at these two wavelengths. With knowledge of $\alpha_{mol,abs}$, it is possible to calculate the concentration of gas atoms or molecules in the investigated substance [93,94]. DIAL systems are effective for atmospheric gases due to their ability to provide continuous high-resolution vertical profiles of gas concentration [91,92].

### 3.5. Doppler (Wind) LiDAR

The Doppler LiDAR leverages the Doppler effect to measure atmospheric wind speed by detecting frequency shifts in the backscattered signal caused by moving particles. Two primary approaches are used to determine principles for calculating the Doppler shift: direct detection (D-DWL) and coherent detection (C-DWL) [95].

D-DWL directly measures the changes in signal intensity or photon counts of the backscattered signal. On the other hand, C-DWL detects both the intensity and the phase and frequency of the backscattered signal [95,96]. C-DWL systems operate by transmitting pulses from a master laser; the backscattered signal—Doppler-shifted by atmospheric motion—is then mixed with a reference signal from the same laser source. This interference produces a beat frequency, which is analyzed to precisely determine the Doppler shift [96]. Compared to D-DWL, coherent detection systems provide higher measurement precision under equivalent signal-to-noise ratio (SNR) conditions, owing to their enhanced capability to resolve narrowband aerosol and cloud backscatter. Additionally, C-DWL systems generally provide superior spatial resolution for wind profilin [95,97].

Doppler LiDARs are widely used in atmospheric research and meteorology to obtain accurate, high-resolution wind measurements of wind profiles. Their application includes tracking weather patterns, assessing air quality, and improving numerical weather prediction models [98].

### 3.6. The Ranging and Altimeter

The concept of laser altimetry is straightforward. It involves measuring the time of flight for a laser pulse to travel to a surface and reflect back to the sensor. Equation (Equation 2) determines the distance *z* from the laser to the surface [99].(2)$$z=\frac{c\Delta T}{2}$$
where *z*[m] is the distance from the laser to the surface. *T*[s] is the time of flight for a laser pulse to travel to a reflective surface and back, and *c*[m/s] is the speed of light in a vacuum = 299,792,458 m/s. The calculated distance, combined with the satellite attitude platform position, leads to the precise three-dimensional spatial coordinates of the laser footprint point.

When combined with the satellite’s geolocation and orientation data, this range information enables the derivation of highly accurate three-dimensional coordinates of the laser footprint on the Earth’s surface. Conventional spaceborne laser altimeters typically operate using high-energy pulses—ranging from 20 to 50 millijoules—and a low pulse repetition frequency (PRF) between 1 and 10 Hz [99].

### 3.7. Full-Waveform LiDAR

Full-waveform LiDAR is a type of LiDAR that records the complete backscattered energy profile of a laser pulse as it interacts with surface and vegetation targets. Unlike the classical altimeter LiDAR systems, which only capture a limited number of return points, full-waveform LiDAR provides a continuous record of the vertical distribution of targets within the laser footprint, enabling detailed characterization of the target such a forest canopy structure, sub-canopy layers, and ground elevation [100,101].

## 4. Challenges and Limitations

LiDAR offers several distinct advantages over both passive and active remote sensing systems. Passive remote sensing techniques require an external illumination source (sun or examined target), which makes them dependent on day/night conditions [18]. Active remote sensing techniques such as SAR or LiDAR have their own source of radiation, making them suitable for day-and-night observation. A key advantage of LiDAR lies in its capability for high-precision vertical profiling, where instruments such as CALIPSO, GEDI, and ICESat-2 provide precise three-dimensional data to measure the vertical structure of clouds, aerosols, forest canopies, and ice sheets. In addition, direct range measurement through flight time can be used to determine heights and layer structures [102]. Notably, Doppler wind LiDAR systems, such as ADM-Aeolus, take advantage of the Doppler shift of scattered radiation to remotely measure wind speeds and temperature profiles, offering a unique capability to capture atmospheric dynamics even under clear air conditions [103]. However, LiDAR systems face several challenges and limitations, which vary across application domains due to differences in system design, operational environments, and performance expectations:Transmitter design challenges: The transmitter laser has every component, including the laser resonator, with elements such as laser crystals, Q-switches, harmonic generator crystals, wave plates, mirrors, and other optical components. These components must meet operational lifetime without worsening performance [24]. One solution to this challenge can be component redundancy; for example, the LITE laser transmitter deployed two identical lasers, and the GLAS laser transmitter uses three identical lasers that do not operate simultaneously [21,104].Spatial resolution: Compared to other remote sensing techniques, LiDARs suffer from low spatial resolution. For example, NASA’s GEDI LiDAR uses 25 m diameter laser footprints spaced 60 m apart along the track (and 600 m across the track) which yields a sparse sampling of the surface rather than a continuous image. In contrast, passive optical satellites can achieve much finer horizontal resolution: commercial imagers like WorldView-3 have pixels as small as 0.31 m [105,106].Spatial coverage: LiDAR is distinguished as having relatively small coverage and swath width. For comparison, Landsat-9 (passive, optical) has a swath area of 185 kilometers (km), covering the whole world every 16 days. Sentinel-1 (active radar) has a swath area of 290 km, covering the whole world every 6 days. GEDI has the widest spaceborne LiDAR swath with 4.2 km, making it possible to cover about 2–4% of the land during its 2-year mission [18,107].Weather dependency: Laser light in the visible to near-infrared spectrum is strongly affected by clouds, rain, and other atmospheric conditions that can scatter or absorb the laser pulses. In contrast, SAR operates in the microwave region, which is largely unaffected by such conditions, allowing it to “see” through clouds and perform reliably in almost any weather [108].Multiple scattering signals: Cloud and aerosol measurements are complicated by multiple scattering phenomena that require complex correction algorithms [109].

## 5. LiDAR in Space Missions

The first spaceborne laser altimetry system, deployed during Apollo missions, marked the beginning of LiDAR development [110,111]. This section provides an overview of key historical milestones and currently operational spaceborne LiDAR instruments, highlighting their contributions to the advancement of LiDAR technology.

### 5.1. Spaceborne LiDARs for Terrestrial Applications

Several spaceborne LiDAR missions have been launched to support Earth observation and environmental monitoring. Table 1 summarizes the main missions, detailing the name of the mission, the responsible agency, the deployment platform, the launch year, the LiDAR instrument, the operational status, and the primary scientific objective. Selected missions are discussed in the following subsections, with a focus on the LiDAR instruments and their contributions to spaceborne remote sensing. Representative images and renderings of the deployment platforms are presented in Figure 5. Table 2 provides a basic parameter about the LiDAR instruments used as a payload in the missions of Table 1.

**Figure 5 sensors-25-03696-f005:**
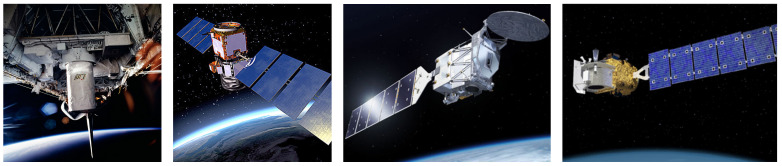
The selected deployment platforms carrying LiDAR instruments, from the left: LITE instrument on top of the Space Shuttle Discovery, CALIPSO satellite carrying a CALIOP instrument, EarthCARE carrying ALTID and ICESat-2 with ATLAS instrument [112,113,114,115].

**Table 1 sensors-25-03696-t001:** A list of missions carrying LiDAR instruments for Earth monitoring.

Mission	Agency	Deployment Platform	Launch Year	LiDAR Instrument	Status	Primary Objective	Cit
STS-64	NASA	Space Shuttle Discovery	1994	LITE	Completed in 1994	Test of the spaceborne LiDAR and its related key technologies, investigate the molecular atmosphere, aerosols and clouds	[7,16]
ICESat	NASA	ICESat	2003	GLAS	Completed in 2009	Measure ice sheet mass balance: the difference between an ice sheet’s snow input, and the ice loss through melting, ablation, or calving	[60,116]
CALIPSO	NASA, CNES	CALIPSO	2006	CALIOP	Completed in 2023	Study the role that clouds and aerosols play in regulating Earth’s weather, climate and air quality	[73,117,118,119]
CATS	NASA	ISS	2015	CATS	Completed in 2017	Extend global LiDAR climate observations, measure range-resolved profiles of atmospheric aerosol and cloud distributions and properties, testing new LiDAR technologies	[120,121]
ADM-Aeolus	ESA	ADM-Aeolus	2018	ALADIN	Completed in 2023	Provide global observations of wind profiles with a vertical resolution that meets the accuracy requirements of the World Meteorological Organization (WMO)	[122,123]
ICESat-2	NASA	ICESat-2	2018	ATLAS	Active	Measure polar ice sheet mass balance, sea ice thickness, and vegetation canopy height better to understand climate change and its impacts	[124]
GEDI	NASA	ISS	2018	GEDI	Paused	Optimized to measure ecosystem structure - determine how changing climate and land-use impact ecosystem structure and dynamics. Measurement of the canopy structure, biomass and topography.	[107]
Daqi-1	CNSA	Daqi-1	2022	ACDL	Active	First HSRL in space. Measure aerosol profiles and greenhouse gas ($CO_{2}$) concentrations	[125]
Goumang	CAST, CRESDA	Goumang	2022	LiDAR	Active	Designed for forest carbon sink observation using both LiDAR and passive sensors. Increase the accuracy and efficiency of carbon dioxide measurements. Detect vegetation biomass, atmospheric aerosols, and chlorophyll fluorescence to view the carbon cycle comprehensively.	[126]
EarthCARE	ESA, JAXA	EarthCARE	2024	ATLID	Active	Observe the vertical profiles of natural and anthropogenic aerosols globally, including their radiative properties and interactions with clouds. Observe the vertical distributions of atmospheric liquid water and ice globally, their transport by clouds, and their radiative impact. Retrieve profiles of atmospheric radiative heating and cooling by combining the retrieved aerosol and cloud properties	[127,128]

**Table 2 sensors-25-03696-t002:** A list of LiDAR instruments carried as payloads on the missions shown in Table 1. H. res stands for horizontal resolution, V. res for vertical resolution, PC indicates photon-counting mode, and AB denotes atmospheric backscatter.

Instrument	Type	PRF [Hz]	No. Lasers	No. Beams	Channels	Laser e. [mJ]	H. Res ^1^ [m]	V. Res ^1^ [m]	Footprint Diameter [m]	Swadth Width	Detector Mode	Detector	Source
LITE	AB	10	2	1	1064	470,440	740	15	290, 470	-	Waveform	APD	[16,20,104]
					532	530,560					Waveform	PMT	
					355	170,160					Waveform	PMT	
GLAS	Altimeter	40	3	1	1064	75	∼170	0.15	∼70	-	PC	APD	[60,116,129,130]
	AB				532	35		78.6			PC	PMT	
CALIOP	AB	20.16	2	1	1064	110	333	30	∼70	-	Waveform	APD	[73,131,132]
					532						Waveform	PMT	
					532						Waveform	PMT	
CATS	AB + HSRL	4000 ^2^	2	1 ^2^	1064	2 ^2^	350	60	∼14.38	-	PC	N/A	[120,133,134]
					532						PC	N/A	
ALADIN	Doopler	50	1	1	355	80	∼87,000	250 m	-	-	N/A	CCD	[135]
ATLAS	Altimeter	10,000	2	6	532	0.2–1.2	0.7	N/A	∼13	6600	PC	PMT	[124]
GEDI	full-waveform	242	3	8	1064	10	60	N/A	25	4200	Waveform	Si:APD	[136,137]
ACDL	HSRL	40	N/A	N/A	1572	N/A	N/A	24	70	N/A	N/A	PMT	[125]
					1064	180							
					532	130							
Goumang	full-waveform	40	N/A	5	1064		-	N/A	N/A	N/A	N/A	N/A	[138]
ATLID	HSRL	51	1	1	355	38	10 km	100,300	N/A	-	-	CCD	[139,140,141]

^1^ The finest achieved resolutions are depicted in the table. ^2^ These values for the CATS LiDAR correspond to its second operational mode, as the first one failed after a few days of operation.

#### 5.1.1. LITE—LiDAR In-Space Technology Experiment

The LiDAR In-Space Technology Experiment (LITE), deployed aboard the Space Shuttle Discovery during the STS-64 mission (9–20 September 1994), was the first spaceborne elastic backscatter LiDAR designed to observe multilayer atmospheric structures, particularly clouds and aerosols, from orbit. The mission validated critical technologies for space-based LiDAR operations, including laser system performance, thermal management, optical alignment, and system [104,142]. This was accomplished by investigating molecular atmosphere, aerosols, clouds, ground properties, atmospheric temperature, and density in the 25 to 40 km altitude range [104,143]. LITE operated for approximately 53 h, collecting around 40 GB of data, significantly contributing to subsequent spaceborne LiDAR developments [7].

The initial raw data immediately revealed the enormous promise of using LiDAR technology in space. Observations included layers of desert dust, biomass burning, continent-level pollution, volcanic aerosols in the stratosphere, and various storm systems [7]. The detailed description of the LITE instrument can be found in [16].

#### 5.1.2. GLAS—The Geoscience Laser Altimeter System

The Geoscience Laser Altimeter System (GLAS), onboard NASA’s Ice, Cloud, and Land Elevation Satellite (ICESat), was launched on 13 January 2003, and operated for approximately 6.5 years in a 600 km polar orbit with a 94° inclination. Its primary objective was to measure ice-sheet mass balance and assess the effects of atmospheric and climate changes on polar ice masses and global sea level trends. GLAS also acquired valuable data on cloud and aerosol distribution, land topography, sea ice conditions, and vegetation cover [21,144].

#### 5.1.3. CALIOP—Cloud-Aerosol LiDAR with Orthogonal Polarization

The Cloud-Aerosol LiDAR with Orthogonal Polarization (CALIOP) was launched aboard the CALIPSO satellite on 28 April 2006, as part of a collaborative mission between NASA (USA) and CNES (France). Operating at an altitude of 705 km, CALIPSO is integrated within NASA’s Aqua satellite constellation, a group of satellites designed to collectively study Earth’s climate system through synergistic data acquisition [88,117,119].

CALIOP was the first spaceborne LiDAR specifically optimized for aerosols and cloud profiling, as well as the first polarization-sensitive LiDAR in orbit. Its primary scientific objective was to precisely measure the altitudes, spatial distributions, and overlaps of the cloud and aerosol layers [131].

#### 5.1.4. CATS—The Cloud–Aerosol Transport System

The Cloud–Aerosol Transport System (CATS) is an elastic backscatter LiDAR instrument launched to the International Space Station (ISS) in January 2015. The Cloud–Aerosol Transport System (CATS) is an elastic backscatter LiDAR instrument launched to the International Space Station (ISS) in January 2015. Operating from a 415 km orbit with a 51° inclination, CATS aimed to provide near real-time aerosol data to support global climate modeling, fill temporal gaps left by missions like CALIPSO, and test new LiDAR technologies. Notably, it evaluated high-frequency lasers, photon-counting detectors, and multibeam systems. The processed aerosol data were delivered within approximately six hours to support the timely forecasting of aerosols [120,133,134].

#### 5.1.5. ATLAS—Advanced Topographic Laser Altimeter System (ICESat-2)

ICESat-2, an altimeter instrument, was launched in September 2018 as a high-priority follow-up to the successful ICESat-1 mission. ICESat-2 aims to measure polar ice sheet mass balance, sea ice thickness, and vegetation canopy height to understand climate change and its impacts. Its ATLAS system also captures atmospheric data, improves climate models, and extends existing data records [46,61,124]. The system itself can operate in two models: (1) backscattering mode to measure the volume properties of the targets such as clouds and aerosols and (2) sounder mode, where the laser pulse TOF is used to measure surface topography, vegetation mass, sea ice thickness, etc. [46,61,124].

Earth observation accuracy and reliability with ICESat-2 were improved compared to ICESat-1, which had a 70 m footprint and a 170 m spacing. In its first two years, ICESat-2 collected global surface data with a density of 70 cm apart in the along-track direction and a maximum spacing of less than 2 km in the vertical track direction at the equator. As mentioned above, a multibeam approach is used to obtain denser spatial sampling [26].

#### 5.1.6. GEDI—The Global Ecosystem Dynamics Investigation

The Global Ecosystem Dynamics Investigation (GEDI) is a joint NASA and the University of Maryland mission launched to the International Space Station (ISS) on 5 December 2018. Mounted on the Kibo Exposed Facility of the ISS, GEDI collects data primarily to assess forest structure and above-ground biomass worldwide. The initial duration of the mission was 48 months, starting 25 March 2019; after an extension until March 2023, it resumed for a second phase on 22 April 2024 [107,136]. Its primary goal is to generate high-resolution laser-based measurements to map forest biomass in temperate, subtropical, and tropical ecosystems. During the original lifetime of the mission, the system was expected to produce approximately 10 billion cloud-free observations [145,146]. During the original lifetime of the mission, the system was expected to produce approximately 10 billion cloud-free observations [146].

### 5.2. Atmospheric Laser Doppler Instrument (ALADIN)

ALADIN, aboard ESA’s ADM-Aeolus satellite launched in 2018, is the first Doppler wind LiDAR designed for space-based direct detection of global wind profiles [26,147]. ALADIN provides vertical wind profile observations meeting World Meteorological Organization (WMO) accuracy requirements, crucial for weather forecasting and climate studies [95,148]. The satellite orbits at a Sun-synchronous altitude of 320 km, with a 7-day repeat cycle [148]. The receiver uses a 1.5 m diameter telescope, combined with two spectrometers: the Mie spectrometer for aerosol and cloud particle backscatter, and the Rayleigh spectrometer for molecular backscatter analysis. Wind profiles are resolved vertically at 250 m intervals from altitudes of approximately 2 km to 30 km [135,149].

#### ATLID—Atmospheric LiDAR (ATLID)

Atmospheric LiDAR (ATLID) is a high-spectral-resolution backscatter Light Detection and Ranging (LIDAR) instrument aboard the EarthCARE satellite. ATLID is designed to detect cloud boundaries and to profile optically thin clouds and aerosols with high precision [141]. EarthCARE itself operates in a sun-synchronous orbit at an altitude of 393.14 km with an inclination of 97.05°. The satellite completes an orbit every 92.5 min and repeats its ground track every 25 days. This orbit ensures consistent lighting conditions for coordinated atmospheric measurements [139]. ALTID provides vertical resolutions of 103 m (ground to 20.2 km) and 500 m (20.2–40 km). Achieving an initial 140 m horizontal sampling resolution along the satellite track. To improve the signal-to-noise ratio and provide more robust data for atmospheric studies, these measurements are averaged onboard to produce a final horizontal resolution of approximately 10 km. A 60 cm telescope collects the backscattered photons, which are processed to distinguish Mie scattering (narrow-band scattering by aerosols and particles) and Rayleigh scattering (broad-band scattering by atmospheric molecules). Co-polarised and cross-polarised signals are separately detected to classify aerosol types [140,141].

### 5.3. Spaceborne LiDARs Beyond Earth

Although this paper primarily covers terrestrial applications, it is essential to acknowledge LiDAR’s pivotal role in extraterrestrial exploration. Table A1, Table A2 and Table A3 in the Appendix A briefly characterize the mission where the LiDAR instrument was used as a payload. It is important to note that this selection includes missions that utilize LiDAR as a scientific instrument. However, there are other missions, such as (e.g., Chang’e-3, Chang’e-4), using short-range LiDAR for precise landing and hazard avoidance rather than for global or regional topography mapping. These missions are not included in Table [150,151].

## 6. Applications and Outcomes from Spaceborne LiDAR Data

Data from spaceborne LiDAR systems have facilitated numerous studies and practical applications in Earth observation, environmental monitoring, and planetary science. This section categorizes key applications, summarizing representative studies and findings.

### 6.1. Atmospheric Applications

Spaceborne LiDAR systems provide unparalleled insight into the vertical structure of Earth’s atmosphere, especially for aerosols (particles) and clouds.

Ref. [152] analyzes the Earth’s boundary layer height (PBLH) and applies the Different Thermo-Dynamics Stability (DTDS) algorithm to CATS spaceborne LiDAR data, allowing for a direct comparison between satellite- and ground-based LiDAR measurements. The study validates PBLH estimates using data from the Atmospheric Radiation Measurement (ARM) program, ensuring consistent detection at all hours. The refined approach aims to improve the ability to obtain the diurnal variability of PBLH from satellite observations.

Ref. [119] utilizes data from the CALIOP LiDAR on the CALIPSO satellite, which has been collecting global atmospheric profiles since June 2006. The research aims to characterize the global 3-D distribution of aerosols, including their seasonal and interannual variations, by constructing a monthly global gridded data set of daytime and nighttime aerosol extinction profiles, available as a Level 3 aerosol product. The data set distinguishes between cloud-free and all-sky conditions and reveals that vertical aerosol distributions vary with season due to changes in source strengths and transport mechanisms. The study finds that the mean aerosol profiles in the clear sky and in the all-sky atmosphere are quite similar, indicating a lack of correlation between high semitransparent clouds and aerosols in the lower troposphere. An initial evaluation of the accuracy of aerosol extinction profiles highlights detection limitations, particularly in the upper troposphere. Despite preliminary results, the study provides evidence that monthly mean CALIOP aerosol profiles can quantitatively characterize elevated aerosol layers in major transport pathways. The work forms the basis for an initial global 3D aerosol climatology, which will be extended and improved over time.

Ref. [153] presents a global and seasonal distribution of cirrus clouds based on measurements from the CALIPSO satellite LiDAR, collected between June 2006 and June 2007. The frequency of occurrence of cirrus clouds is highest near the tropics, particularly in the 100° to 180° E longitude band, with a maximum frequency of up to 70%. Cirrus cloud cover shows significant latitudinal movement with changing seasons. The vertical distribution reveals a maximum frequency of occurrence of the highest altitude of the cirrus cloud at approximately 11% at 16 km in the tropics. In the northern and southern midlatitudes (20° N to 60° N), the maximum frequency of the top and base altitudes of the cirrus is about 5.1% at 11 km and 8 km, respectively. The study highlights the variability in the distribution of the cirrus cloud horizontally and vertically, influenced by seasonal changes.

### 6.2. Vegetation and Ecosystem Monitoring

Data produced by spaceborne LiDARs allow us to observe Earth’s vegetation, especially forest structure [154] and biomass. Forest ecosystems have been one of the most extensively studied applications of spaceborne LiDAR. Missions like NASA’s GEDI provide detailed vertical profiles of the forest canopy, allowing estimation of canopy height, biomass, and structural complexity. Such data contribute to a better understanding of carbon cycling and biodiversity [154,155,156]. An example of studies can include the following.

Ref. [157] investigates the effectiveness of European Protected Areas (PAs) in conserving the vertical structural complexity of forests compared to non-protected forests. The study utilizes over 30 million observations from the GEDI mission. The analysis compares various forest structural metrics, such as canopy top height, Foliage Height Diversity (FHD), and Relative Height (RH) metrics (e.g., RH50 and RH25), inside and outside nearly 10,000 PAs. The findings reveal that forests within PAs are generally taller and more vertically complex than those in unprotected areas, highlighting the positive impact of environmental policies. The study underscores the utility of spaceborne LiDAR for large-scale monitoring of forest attributes essential for conservation and restoration efforts.

The study in [45] aims to assess the applicability of GEDI data to characterize the structure of mountain forests. Comparison of GEDI data with Airborne Laser Scanning (ALS) data to estimate key forest structural parameters on the plot scale, considering the trade-off between data density and seasonal/phenological mismatches. The impact of topography on the estimation of structural parameters is also evaluated. Additionally, the study investigates whether GEDI’s sampling-based approach can capture forest structure variability at the landscape scale, particularly focusing on rare cases. The effectiveness of GEDI data in educating indicators of ecosystem function, such as avalanche protection, carbon storage, and habitat quality, is assessed in two contrasting mountain landscapes in Germany (Berchtesgaden) and Switzerland (Davos). The study finds that the availability of GEDI data varies between the phenological seasons, with a higher availability during the leaf-on season. The agreement between the GEDI and ALS metrics is highest when the data recording dates are closely aligned, with varying levels of agreement between the different metrics.

Ref. [158] leverages 2005 data from GLAS aboard ICESat to create a global map of the height of the forest canopy at a spatial resolution of 1 km. Despite the sparse coverage of the LiDAR shots, the researchers modeled the global canopy height (RH100) by correlating the GLAS-derived values with ancillary variables such as forest type, tree cover, elevation, and climatology maps. The model’s accuracy was validated with field measurements from 66 FLUXNET sites, showing a conservative error margin due to measurement uncertainty and sub-pixel variability. The study highlights a global latitudinal gradient in canopy height, increasing towards the equator, and identifies coarse forest disturbance patterns. The findings underscore the challenges in mapping tall canopies, particularly in closed broadleaf forests such as the Amazon.

Ref. [159] investigates the applicability of GEDI data to characterize tropical forest aboveground biomass (AGB) on the scale required for REDD+ projects. GEDI, a spaceborne LiDAR instrument on the ISS, provides high-confidence measurements to estimate AGB but does not collect data systematically. The study examines the amount of GEDI data needed to reliably assess AGB forest in tropical Africa, specifically in Mai Ndombe province, Democratic Republic of the Congo. Using good quality GEDI footprint-level AGB data for 31 months, the study analyzes 15 test sites with >80% forest cover. It finds that observation periods ranging from 143 to 534 days are required to meet the IPCC accuracy requirement of ±10% for forest AGB estimates. The findings highlight the importance of sufficient data collection periods for effective REDD+ project monitoring and the variability of required observation periods based on local conditions.

### 6.3. Climate Change and Cryospheric Monitoring

Monitoring ice sheets and glaciers can provide insight into the rise in sea levels. Spaceborne LiDAR altimetry provides direct measurements of ice surface elevation and changes over time. These measurements are used to infer mass balance and to monitor the dynamics of ice caps and glaciers. Recent studies using LiDAR for snow and ice measurements underscore its ability to capture even small changes in surface elevation, which are useful climate models for assessing the impact of global warming on polar regions. NASA’s ICESat (2003–2009) and ICESat-2 (2018–present) missions have been instrumental in this regard.

Based on ICESat and/or ICESat-2, several studies were published. For example, Ref. [160] shows that between October 2018 and April 2021, satellite data from ICESat-2 and CryoSat-2 revealed a decrease in the depth and thickness of the Arctic sea ice snow. The mean snow depth in April decreased by approximately 2.50 cm and the ice thickness decreased by approximately 0.28 m, leading to a 12.5% loss in ice volume. Multiyear ice showed significant thinning, with a 16.1% reduction in thickness by 2021, while first-year ice remained relatively unchanged. Using climatology-based snow depth resulted in thicker ice estimates. The study in [161] focuses on understanding the mass loss of Earth’s ice sheets and the climate processes responsible for it. Using satellite laser altimetry data from NASA’s ICESat and ICESat-2 satellites, the study estimates changes in grounded and floating ice mass for the Greenland and Antarctic ice sheets from 2003 to 2019. According to the study, Greenland lost 200 ± 12 gigatons per year, while Antarctica lost 103 gigatons per year, with 118 ± 24 gigatons per year from grounded ice and a small net gain of 15 ± 65 gigatons per year from ice shelves. Together, these losses contributed approximately 14 mm to the global rise in sea level during the 16-year period (8.9 mm from Greenland and 5.2 mm from Antarctica).

### 6.4. Bathymetry—Deriving Underwater Topography

Bathymetry, the science of measuring and mapping underwater topography, is essential to understanding coastal dynamics, marine habitats, and navigation safety. NASA’s Ice, Cloud, and Land Elevation Satellite-2 (ICESat-2), equipped with the Advanced Topographic Laser Altimeter System (ATLAS), uses green-wavelength laser pulses to penetrate clear coastal waters and measure seafloor depths of up to 40 m under optimal conditions. Although bathymetry was not part of the original objectives of ICESat-2’s mission, the satellite has been proven to be capable of providing valuable bathymetric data, particularly in the coastal and near-shore regions [162,163]. Multiple recent studies utilizing ICESat-2 data have been published:

Ref. [164] describes the creation and evaluation of ATL24, a global bathymetric data product for coastal and nearshore measurements provided by ICESat-2. The team analyzed the mapping performance of the data set and validated its precision at eight different sites. The findings help researchers understand the bathymetric capabilities of ICESat-2 and guide the use and future improvement of ATL24 for scientific and practical applications.

Ref. [162] provides a comprehensive review of the algorithms used to extract bathymetry data from ICESat-2 satellite measurements. It covers key steps including water surface detection, bottom return classification, refraction correction, accuracy evaluation, and integration with optical imagery. The aim is to guide the development of a standardized global bathymetry product (ATL24) and assist researchers in refining and testing new algorithms for nearshore mapping.

Ref. [165] presents a new method to measure the depth of Arctic supraglacial lakes using ICESat-2 and Sentinel-2 satellite data, overcoming the limitations of traditional methods. By combining spectral analysis and a refined regression model, the method improves the accuracy of the depth estimation. The tests in Greenland lakes showed error reductions of up to 14%, which improved the ability to monitor lake volume changes and provided better information on the impacts of climate change.

## 7. The Future of the Spaceborne LiDARs

In this section, the future direction of spaceborne LiDARs is described. Starting with planned missions that are planned to hold a LiDAR as a scientific instrument, we continue with the currently developed concept.

### 7.1. Future Missions

The next decade promises advances in spaceborne LiDAR, with multiple agencies planning new missions and technological improvements on the horizon. Several upcoming missions are in development:

#### 7.1.1. MERLIN

An eagerly anticipated mission is the MERLIN satellite (a collaboration of CNES and DLR) due for launch around 2028, which will carry a LiDAR to measure atmospheric methane concentrations. MERLIN will use an Integrated Path Differential Absorption (IPDA) LiDAR to map methane with 50 km resolution, operating day and night at high latitudes where passive sensors have limitations. Its success could pave the way for a suite of greenhouse gas LiDARs (NASA has considered a similar $CO_{2}$ IPDA concept under the name ASCENDS) [166,167,168].

#### 7.1.2. AEOLUS2

Regarding the wind measurement, there are discussions within ESA of a follow-up operational mission of Aeolus, potentially Aeolus-2. This mission would deploy multiple Doppler LiDARs to provide continuous global wind data for weather forecasting, essentially moving from a demo to an operational system. Such a system might address Aeolus’s weaknesses by using improved lasers (maybe in the UV or even eye-safe bands) with longer lifetimes and possibly two satellites for morning/evening orbit coverage [169,170].

#### 7.1.3. Multi-Footprint Observation LiDAR and Imager (MOLI)

JAXA’s Multi-Footprint Observation LiDAR and Imager (MOLI) aims to map the forest canopy and evaluate biomass. MOLI will employ a two-beam LiDAR system that creates parallel paths separated by 50 m, allowing detailed canopy surface measurements. Operating at a wavelength of 1064 nm with a pulse repetition frequency of 150 Hz, MOLI is designed to provide high-resolution data with a 25-meter resolution. This instrument, which will be deployed on the International Space Station (ISS) from 2024 to 2028, aims to improve our understanding of forest structure and contribute to global environmental monitoring efforts [171,172,173].

#### 7.1.4. Gualan

The Guanlan Science Mission is a Chinese initiative that aims to advance space oceanography. It will combine two key technologies: interferometric altimetry (IA) and ocean LiDAR (OL). Using dual-frequency (Ku and Ka bands), the IA will provide high-resolution ocean surface topography measurements. The OL, the first spaceborne active LiDAR designed for oceanography, will penetrate deeper into the ocean to profile optical properties and detect marine life in the euphotic layer. This mission promises to improve our understanding of ocean dynamics, bio-optical properties, and marine ecosystems, ultimately contributing to global ocean observation efforts [174].

### 7.2. Future Contepts

#### 7.2.1. Quantum LiDAR

Quantum-enhanced LiDAR techniques have the potential to surpass classical noise limits and offer improvements in both range resolution and sensitivity. Although still in the research phase, the use of quantum phenomena such as entanglement can improve detection accuracy and signal robustness, even in low-reflectivity or high-noise environments. In addition, quantum illumination protocols that include entangled photon pairs were shown to provide a quantum advantage in detection probability compared to classical strategies, especially in scenarios dominated by background noise and loss [175].

Recent theoretical advances suggest that quantum LiDAR systems employing squeezed states and frequency-entangled photon pairs could achieve enhanced precision in target velocity estimation by reaching the Heisenberg limit [176]. Additionally, the resilience of certain quantum states against photon losses offers a substantial advantage over classical limited coherent state protocols. The utilization of multimode squeezed vacuum states in quantum Doppler LiDAR has been shown to enable better velocity estimation [176].

However, the practical realization of a Quantumn LiDAR is complex. It necessitates the development of highly efficient entanglement sources and low-loss quantum memories to preserve the idler photon over extended durations. Moreover, the optimized receiver designs can extract the full quantum advantage [175]. As quantum technologies evolve, future spaceborne LiDAR systems may leverage these advancements to achieve superior performance while operating at lower power budgets.

#### 7.2.2. Swath Mapping

The transition from single-beam to swath-mapping LiDAR, facilitated by the use of pixelated detectors, represents an advancement in remote sensing technology. Using coverage with tens to hundreds of simultaneous measurements, the LiDAR swath mapping vastly improves spatial resolution, improving the ability to capture fine surface features and morphological variations [2]. Traditional single-beam systems rely on discrete pulses to map topography at predefined intervals. However, swath-mapping LiDAR operates by distributing an array of laser beams across a wide field of view. This transition can bring about improvements in planetary science and Earth observation. It allows the generation of high-resolution topographic data sets and can be helpful for applications such as geodetic control, surface roughness analysis, and tidal deformation studies. However, the implementation of swath-mapping LiDAR requires careful consideration of power constraints, because increasing the number of simultaneous measurements often necessitates higher laser power to maintain sufficient signal-to-noise ratios [2].

#### 7.2.3. LiDAR Sattelite Constalations

Deployment of a constellation of satellites with LiDAR payloads enables near-continuous, global data collection, allowing for more frequent and timely updates of Earth’s surface conditions. An Orlando startup named NUVIEW plans to develop and launch the first constellation of LiDAR-equipped satellites, counting 20 satellites in total, promising 100× faster data collection than conventional methods [177].

#### 7.2.4. LiDAR as a CubeSat Playload

Recent developments in miniaturization have made it possible to equip CubeSats with LiDAR systems, opening new opportunities for affordable and frequent 3D Earth observation. These small satellites offer a cost-effective alternative to large, traditional missions, while still providing valuable data for applications like terrain mapping, forest monitoring, and atmospheric research. For example, Ref. [178] describes the design and analysis of a CubeSat equipped with a LiDAR sensor for environmental monitoring. Ref. [179] explores various LiDAR concepts for space applications using CubeSats. It covers short-range proximity LiDARs for 3D imaging in satellite rendezvous and sample capture missions, and long-range configurations (up to 1000 km) for planetary mapping and altimetry.

## 8. Conclusions

Spaceborne LiDAR has come a long way since its inception in the 1960s. The tens of missions developed and launched by multi-world agencies (NASA, ESA, CNES, JAXA, …) carrying LiDAR instruments have been developed to explore our planet and beyond. Missions like ICESat-1 and ICESat-2 have contributed to a better understanding of the balance of the polar ice mass. GEDI has measured forest structure and the distribution of above-ground biomass in the tropics and temperate regions. Other instruments, such as CALIOP on the CALIPSO satellite, have provided data on the vertical distribution of clouds and aerosols, enabling deeper insight into climate and atmospheric dynamics. In the future, missions like MERLIN, AEOLUS2, MOLI, or Gualan are planned in order to follow their successors and bring insights into unexplored areas of our planet and beyond.

## Figures and Tables

**Figure 1 sensors-25-03696-f001:**
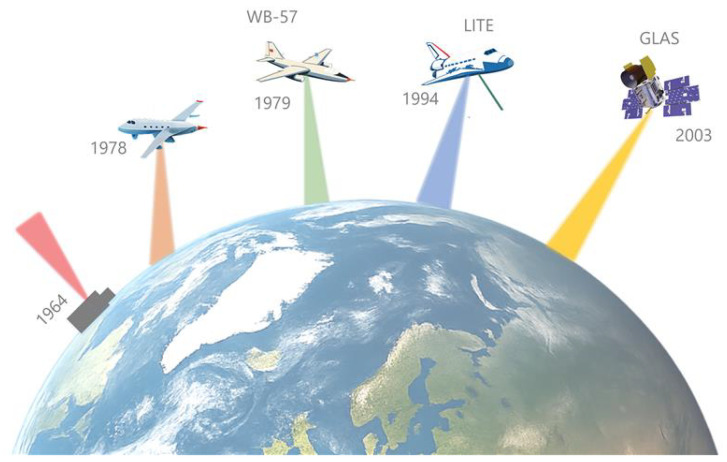
An artistic visualization of the evolution of LiDAR systems based on a deployment platform constructed for Earth exploration, with important milestones depicted inspired by [7]: 1964—the first ground-based LiDAR, atmospheric backscattering LiDAR located in at Lexington, Massachusetts [19]; 1979—the first high attitude airborne LiDAR mission on Wb-57 aircraft [7]; 1994—LITE experiment, the first spaceborne LiDAR (atmospheric backscattering), deployed on Space Shuttle Discovery [20]; 2003—GLAS, the first standalone spaceborne LiDAR [21].

**Figure 2 sensors-25-03696-f002:**
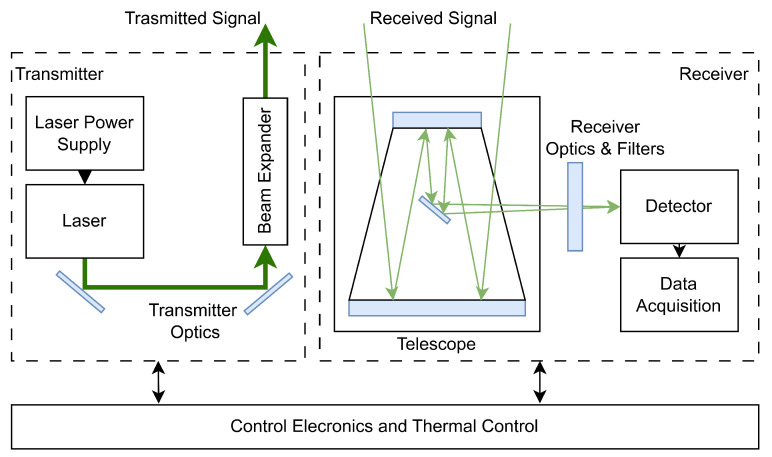
Schematic diagram of the LIDAR instrument architecture. The system includes a transmitter with a laser and beam expander, a telescope for signal collection, and a receiver equipped with optics, filters, a detector, and data acquisition components. The control electronics and thermal control unit manage system operations and maintain temperature stability.

**Figure 3 sensors-25-03696-f003:**
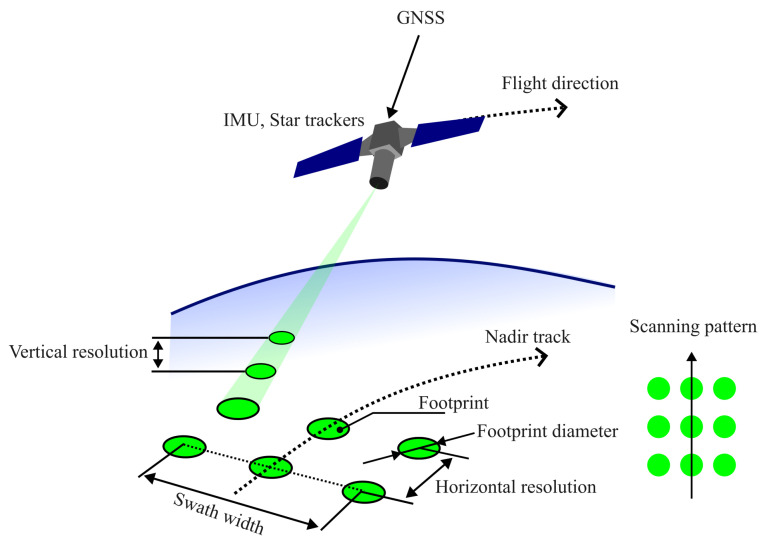
Measurement geometry of a spaceborne LiDAR system. The satellite platform has GNSS, IMU, and star trackers for precise geolocation. The emitted laser pulses generate footprints on the ground, defining the system’s horizontal and vertical resolution. The scanning pattern and swath width determine the spatial coverage of the measurements.

**Figure 4 sensors-25-03696-f004:**
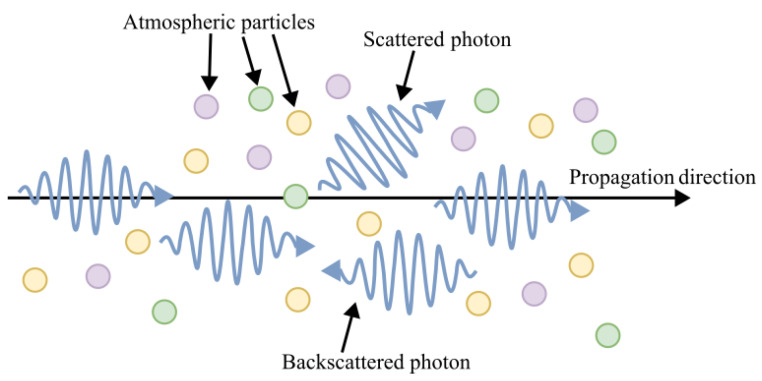
Schematic illustration of photon scattering in the atmosphere. As the laser pulse propagates through the air, photons interact with atmospheric particles, resulting in scattered and backscattered photons. The backscattered photons are captured by the LIDAR receiver and used to infer atmospheric properties.

## Data Availability

All data are contained within the paper.

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
