# Peer review of "Spaceborne LiDAR Systems: Evolution, Capabilities, and Challenges"

_sensors, 2025, doi:10.3390/s25123696_

Round 1
Reviewer 1 Report
Comments and Suggestions for Authors
This is an interesting paper on satellite-based lidar systems. The authors assemble a lot of technical information on the various platforms and provide an interesting history of space-based lidar systems. However, the paper is very long, and in some sections reads more like a catalog of systems rather than an evaluation of their respective merits, deficiencies, and applications. I’d suggest better focusing the paper by moving a lot of the material in section 5 to a supplement, and instead provide a shorter summary of the various systems and their applications. Similarly, the inclusion of extra-terrestrial lidar systems (section 5.6) is a bit off the focus of the paper it seems to me, except if the authors can link capabilities of some of these systems to potential applications to earth-focused systems. Additionally, the authors miss a growing body of research on application of space-based lidar, particularly ICESAT2, to measuring bathymetry and water levels (see Parrish, et al. 2022). Finally, the inclusion of the section on Archaeology (6.4) seems out of place since most of those applications are with aerial or drone-based systems and not space-based systems due to the resolution.
Specific comments:
line 14: You might specify different spellings of LiDAR (e.g. lidar, lidar, etc.)
line 33: DEMs have been important in earth science long before the advent of lidar, although obviously generated from other techniques (photogrammetry, radar, etc.)
line 83: should be “notably” rather than notable.
line 97-101: Perhaps this level of technical detail on Q-switching is unnecessary? Does this add significantly to the understanding of these systems?
Line 151: should be “referred to” rather than referenced.
line 222: should be “LiDAR targets”
lines 246-252: How useful are these resolutions compared to aerial lidar for environmental applications?
Line 299-327: Is this level of detail on scattering necessary to understand these systems?
Lines 546-547: What is the vertical resolution of these systems? Important to understand limitations to capabilities.
Line 700: Should the statement “base altitudes of the cirrus is about 5.” - should that be 5%?
Line 719: should read “The study in [96]”
Reviewer 2 Report
Comments and Suggestions for Authors
Please see the comments in details below:
- Table-1, Please give the reference or citation for each listed spaceborne lidar.
Table-1, please list the lidar type for eachc listed spaceborne lidar as possible.
- Line 218, Section 2.1.3. Data acquisition: No any content?
- Equation-1, lack of R2 in the right side of Equation.
- Line 188, what is the means for “linear mode silicon APDs”?
- Section 3.4, any spaceborne DIAL.
- Section-4: Challenges and limitations. When discussing the limitations and challenge, it might be better to separate Atmospheric application Lidar and Terrestrial lidar (surface elevation, vegetation, and ocean-geography, etc.), because the lidar systems for the Atmospheric and Terrestrial applications are much different on technology and designs.
- Line 511-512, the description on the CALIOP transmitter “two orthogonal polarization components at 532 nm wavelengths (parallel and perpendicular to the polarization plane of the transmitted beam).” It is confused. Please check it.
- Line 826, “1.4. Gualan” is misspelled. Please double-checked it.
Reviewer 3 Report
Comments and Suggestions for Authors
Overall Evaluation
This review presents a comprehensive review of spaceborne LiDAR systems, discussing their evolution, instrumentation, operating principles, mission applications, and future directions. The paper is well-structured, technically sound, and supported by extensive references.
Strengths
-
Thorough and Systematic Content: The review covers all relevant aspects including laser transmitters, detectors, LiDAR types, mission histories, and technological challenges.
-
Comprehensive Referencing: The manuscript draws on a wide range of scientific missions and technological developments, enhancing its credibility.
-
Relevance and Impact: The topic is highly relevant to current and future Earth observation and planetary exploration missions, making it valuable for both academic and applied research communities.
Minor Technical Corrections
-
The term “Doopler” should be corrected to “Doppler” throughout the manuscript.
-
Equations (such as in DIAL section) should include a symbol list and units for clarity.
-
Units in figures and text (e.g., mrad, µrad) should be standardized.
Major Suggestions for Improvement
-
Language Refinement
-
Several sections, especially in the Introduction and Section 2, could benefit from concise language and removal of redundancy.
-
The manuscript would benefit from a thorough linguistic polish to meet the standard of a high-quality review article.
-
-
Figures and Illustrations
-
Figures such as Figure 1, 2, and 4 lack clear captions and high resolution. All images should have well-explained legends and properly referenced sources.
-
Figure 3 could be improved by providing definitions and explanations for the annotated parameters.
-
-
Redundancy and Incomplete Sections
-
Section 2.1.3 “Data acquisition” is listed but not elaborated upon. Please either complete or remove it.
-
Some descriptions in Section 3 (e.g., LiDAR categories) overlap with earlier sections and could be integrated or reduced.
-
-
Future Prospects (Section 7)
-
The discussion on future trends is relatively brief. The authors are encouraged to expand on advanced directions such as miniaturized LiDARs, AI-enhanced data analysis, or quantum LiDAR technologies.
-
-
Formatting and Citations
-
Reference formatting is inconsistent (e.g., numbered references should be enclosed in brackets [ ] and consistently formatted throughout).
-
Some in-text references appear without clear sources or links; these should be clarified and verified.
-
-
-
Round 2
Reviewer 3 Report
Comments and Suggestions for Authors
This manuscript has fully considered my suggestions and I think it can be accepted.